# Single-molecule force spectroscopy of toehold-mediated strand displacement

Andreas Walbrun [1,4], Tianhe Wang[2,4], Michael Matthies [2], Petr Šulc [2,3], Friedrich C. Simmel [2] ✉ & Matthias Rief [1] ✉

Toehold-mediated strand displacement (TMSD) is extensively utilized in dynamic DNA nanotechnology and for a wide range of DNA or RNA-based reaction circuits. Investigation of TMSD kinetics typically relies on bulk fluorescence measurements providing effective, bulk-averaged reaction rates. Information on individual molecules or even base pairs is scarce. In this work, we explore the dynamics of strand displacement processes at the single-molecule level using single-molecule force spectroscopy with a microfluidics-enhanced optical trap supported by state-of-the-art coarse-grained simulations. By applying force, we can trigger and observe TMSD in real-time with microsecond and nanometer resolution. We find TMSD proceeds very rapidly under load with single step times of 1 μs. Tuning invasion efficiency by introducing mismatches allows studying thousands of forward/backward invasion events on a single molecule and analyze the kinetics of the invasion process. Extrapolation to zero force reveals single step times for DNA invading DNA four times faster than for RNA invading RNA. We also study the kinetics of DNA invading RNA, a process that in the absence of force would rarely occur. Our results reveal the importance of sequence effects for the TMSD process and have relevance for a wide range of applications in nucleic acid nanotechnology and synthetic biology.

Nucleic acid strand displacement (SD) occurs whenever two DNA or RNA strands with the same or similar sequences attempt to bind to a complementary target strand, which plays a vital role in many biological processes[1], including DNA recombination[2], CRISPR-based target recognition[3,4], and RNA-based gene regulation[5,6]. In a typical SD reaction, an incumbent strand is initially bound within a target duplex. When the duplex temporarily frays and thus exposes unbound nucleotides at its ends, a single-stranded invader can attach, resulting in the formation of a three-stranded intermediate complex. The invader and incumbent strands then compete for binding to the target in a branch migration process[7]. Strand displacement reactions can be biased in one direction by extending the sequence of the target strand

by a short single-stranded region - the so-called toehold - which allows an invader to bind and initiate strand displacement from there. In toehold-mediated strand displacement (TMSD) reactions, the invader is longer than the incumbent and can thus form a larger number of base pairs with the target, ultimately always displacing the incumbent.

TMSD has been extensively employed in nucleic acid nanotechnology[8], where it was used to switch DNA-based molecular assemblies between different conformations, drive molecular machines[7], or perform computations in chemical reaction networks[9,10]. Merging concepts from DNA nanotechnology and synthetic biology, TMSD processes were also utilized to switch functional RNA molecules in vivo, resulting in conditional CRISPR guide

[1]Technical University of Munich, TUM School of Natural Sciences, Department of Bioscience, Center for Functional Protein Assemblies (CPA), Garching, Germany. [2]Technical University of Munich, TUM School of Natural Sciences, Department of Bioscience, Garching, Germany. [3]School of Molecular Sciences and Center for Molecular Design and Biomimetics, The Biodesign Institute, Arizona State University, Tempe, Arizona, USA. [4]These authors contributed equally: Andreas Walbrun, Tianhe Wang. ✉e-mail: simmel@tum.de; matthias.rief@mytum.de

RNAs[11–14] and programmable RNA-responsive riboregulators[15–18] with excellent ON/OFF ratios.

The kinetics of strand displacement reactions have so far been mainly studied using bulk methods or computational modeling. Early studies investigated branch migration in recombination intermediates observed in bacteriophage T4[2]. The kinetics of single-strand branch migration was initially studied using long radioactively labeled DNA fragments[19–21] and later by Förster resonance energy transfer (FRET) using fluorescently labeled oligonucleotides[22,23].

Bulk FRET experiments have also been extensively applied in the context of DNA nanotechnology, where they were used to investigate the influence of toehold length, sequence mismatches, and secondary structure on the kinetics of TMSD reactions[23–26]. Recently, bulk measurements have also been applied to the study of RNA[27] and DNA-RNA hybrid TMSD[28,29] and have found differences between RNA invading a DNA duplex and DNA invading an RNA duplex, as well as sequence-dependent effects on the kinetics. Bulk methods, however, do not allow direct observation of the strand displacement process itself, and information on the process has to be inferred from the kinetics of the overall reaction, which includes the initial binding of the invader to the target.

Single-molecule methods provide complementary information on the reaction kinetics of individual molecules, which is impossible to access in bulk experiments whose observables necessarily are averaged values. In particular, single-molecule force spectroscopy (SMFS) experiments using optical[30] or magnetic[31] tweezers have yielded a better understanding of the folding pathways of proteins and nucleic acids and their functions[32] and have been extensively used to characterize the force dependence of molecular machines such as polymerases[33] or ribosomes[34,35] acting on them. SMFS has been applied to study the sequence-dependent folding of RNA hairpins[36–38] and riboswitch aptamers[39], providing detailed insights into their folding free-energy landscapes and the role of mismatches during folding. Single-molecule supercoiling experiments with magnetic tweezers were used to study R-loop formation by CRISPR-Cas nucleases acting on DNA target duplexes[4,40], which enabled a quantitative description of the target recognition process. Magnetic tweezers were also used to study the dynamics of strand displacement of short oligonucleotides from a closing DNA hairpin under strain[41].

In the present study, we utilized an optical trap setup to investigate the toehold-mediated invasion of DNA or RNA molecules into the stem of nucleic acid hairpin structures containing a complementary sequence via SMFS. Compared to TMSD experiments involving three strands (invader, incumbent, and target strands), incumbent and target sequences are intramolecularly linked to each other via the hairpin loop (Fig. 1A). Invasion of the hairpin stem, thus leads to an unfolding of the hairpin structure but does not result in the release of the incumbent, which allows us to observe multiple invasion, unfolding and refolding processes with the same molecule.

Toehold-mediated invasion of a hairpin such as that studied here underlies the mechanism of recently developed toehold switch riboregulators[15–18], which contain an unpaired toehold sequence at the 5′ end followed by a strong hairpin-like secondary structure that firmly sequesters a ribosome binding site (RBS). In the absence of cognate RNA invader molecules - termed "triggers" in this context - translation from such an mRNA is inhibited. When trigger RNA is present, it can attach to the toehold region and initiate a strand displacement process, which opens the RNA hairpin structure, exposes the RBS, and thus activates the translation of the mRNA. Several design features of toehold switches - length of the stem, size of the loop, presence of mismatches and bulges - have been systematically studied and were found to profoundly affect the performance of these synthetic gene switches[15,18,42,43]. In the following, we study hairpin structures that contain the same 14 nt toehold sequence as that used for a toehold switch developed in ref. [15] In contrast to the original toehold switch, we chose to adapt the sequence by removing the interior bulge, introducing dedicated mismatches into the invader strand, and elongating the stem region of the hairpin. These modifications facilitate direct observation of the invasion process in SMFS. In combination with coarse-grained simulations using OxDNA[44,45], our studies reveal details of the strand invasion process with close to single nucleotide resolution for DNA-DNA, RNA-RNA as well as RNA-DNA strand invasion and thereby expose the remarkable sequence dependence of its dynamics.

## Results
### Force-induced unfolding of DNA and RNA hairpins observed by SMFS
In order to observe strand invasion into nucleic acid hairpins via SMFS, we generated molecular constructs that allowed us to connect the hairpins to 1 µm-diameter silica beads that could be trapped by the two infrared laser beams of a commercial optical trap setup (Fig. 1A, C-Trap® LUMICKS, see Methods for more details). Our toehold hairpins were folded from 124 nt long DNA or RNA sequences derived from a gene regulatory toehold switch[15] and consisted of a 14 nt long single-stranded toehold and a 52 bp long stem connected by a 6 nt long loop

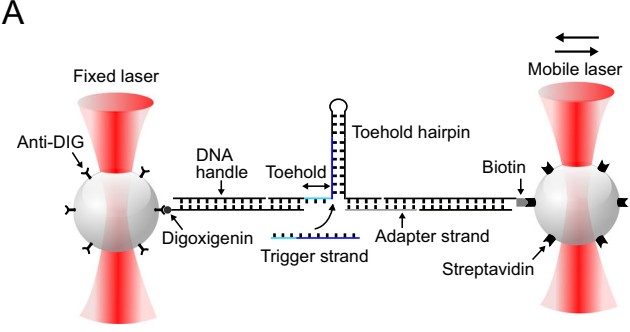

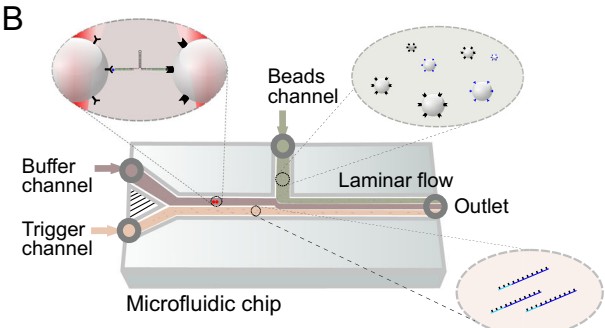

**Fig. 1 | Setup and measurement principle. A** A dual-beam optical trap setup is used to measure force-dependent hairpin opening promoted by a trigger strand in a toehold-mediated strand displacement (TMSD) process. DNA (or RNA) toehold hairpin molecules are tethered between two beads using 545 base pair (bp) long DNA handles as indicated. **B** Microfluidics setup used for TMSD measurements. The hybridized constructs with attached beads are incubated and then pumped into the microfluidics device. The bead mix is introduced into the beads channel together with the pure buffer channel (20 mM MgCl$_2$, 300 mM KCl, 50 mM HEPES)

and trigger channel with the same buffer containing the 100 nM trigger strand. The pure buffer and the buffer containing the trigger strand are separated by laminar flow. The bead pairs are trapped within the beads channel and then transferred to the buffer channel, where two beads are brought into close proximity to form a tether between the DNA handle and anti-digoxigenin beads. Once the tether is formed, pulling and passive mode experiments are performed in the buffer channel, and subsequently, TMSD measurements are executed in the trigger channel.

region (see Supplementary Fig. 1). As schematically shown in Fig. 1A, in a so-called dumbbell assay, the 5′ and 3′ ends of the hairpins were hybridized to two 185 nm long double-stranded DNA handles connecting the hairpin to the silica beads via biotin/streptavidin and digoxigenin/anti-digoxigenin linkers. Since the DNA handles had their overhangs on the 5′ end, a single-stranded adapter (gray in Fig. 1A) was used to hybridize the 3′ end of the hairpin to the handle. Single-molecule experiments were performed using a microfluidic flow cell where a single molecule suspended between the two traps can be exposed to different solution conditions by moving the traps into different microfluidic streams (Fig. 1B). In a typical experiment, the two beads are trapped in the beads channel, the dumbbell is formed in the buffer channel and subsequent measurements can be performed in either a channel containing only buffer solution or in the trigger channel containing 100 nM trigger strand (for more details see Methods section "Measurement preparation").

To characterize the mechanics of the hairpin, we initially performed single-molecule unzipping experiments on both the DNA and the RNA hairpin in the absence of trigger molecules. Stretch/relax cycles obtained at a pulling velocity of 0.2 µm/s of the hairpin are shown in Fig. 2A, B. During stretching, rapid transitions marking increasing unfolding of the hairpin can be observed. Several intermediates (I1, I2, and I3) are populated on the way from the fully folded (Fol) to the fully unfolded (Unf) state. We estimated the number of nucleotides unfolding in the transition from each state to the next by fitting the traces with a worm-like chain (WLC) polymer model[46,47] (colored lines in Fig. 2A, B and Supplementary Table 1). While unfolding forces are significantly higher for RNA as compared to DNA, the length changes from one intermediate to the next are identical in the DNA and RNA hairpins within the resolution of our experiment. Sequence details of the folded and unfolded portions of the partially unfolded intermediates based on our length measurements can be found in Supplementary Fig. 2. Upon relaxation, the molecule readily folds back to the fully folded state (light gray traces in Fig. 2A, B).

To obtain a more detailed kinetic and energetic characterization of the unfolding-refolding equilibrium transitions, we also performed so-called passive mode experiments where we kept the distance between the laser foci constant while observing the fluctuations of the molecule through its intermediate states over tens of seconds (Supplementary Fig. 2C, D). For DNA, a zoom into the data shows 5 different populated levels corresponding to the intermediates mentioned above (Fig. 2A). Assignment and coloring of the states were done using hidden-Markov-modeling (HMM)[48] (see Supplementary Methods "Analysis of SMFS data"). From passive mode data, we calculated the folding free energy of the hairpin from the ratio of the population probabilities of the folded and unfolded state, correcting for energetic contributions from stretching the linkers and spring energies from the beads deflected from the trap centers (see Supplementary Methods "Extracting free energies" for details). We find a folding free energy of $-92.8\ k_BT$ ($-54.9$ kcal/mol), which is in reasonable agreement with the values predicted by nucleic acid thermodynamics software packages (NUPACK[49] (DNA: $-118.88\ k_BT$ ($-70.50$ kcal/mol), mFold[50]: $-118.93\ k_BT$ ($-70.53$ kcal/mol)). The deviations can be explained by systematic errors in the force calibration of the tweezers (see Supporting Information paragraph after Supplementary Table 2 for more details). The kinetics of RNA folding/refolding is significantly slower as compared to DNA (Supplementary Fig. 2D and F). This slow kinetics precludes observation of RNA unfolding at equilibrium. While equilibrium transitions can be observed between intermediates Fol, I1, I2, and I3, the construct stays permanently unfolded as soon as the fully opened state is reached (see sample trace in Supplementary Fig. 2D). Supplementary Table 2 summarizes the free energy values obtained in our experiments compared to calculations using software packages.

## Single-molecule observation of toehold-mediated strand displacement (TMSD)

Here and in the following, we use a nomenclature for the experiments, where the first letter indicates the nature of the invaded hairpin (R: RNA, D: DNA), the second letter denotes the invader (R/D), the third letter denotes the position of a mismatch (p: proximal, c: central), and the following number gives the number of mismatches.

To perform single-molecule TMSD experiments with our setup, we took advantage of the microfluidic capabilities of the optical trap. A toehold hairpin was held at a constant force low enough that the hairpin would be in the folded state but high enough to produce a measurable signal upon strand invasion. When the molecule was transferred from the buffer channel to a channel containing 100 nM of trigger strand molecules (Fig. 1B), binding of a trigger strand to the toehold hairpin and the following invasion of the hairpin stem manifested itself in a sudden drop in trapping force due to the elongation of the invaded hairpin (Fig. 2C, upper trace). For the fully complementary trigger strand sequence, the binding of the trigger strand and completion of the invasion process occur almost simultaneously, and we cannot distinguish between the two events. When we utilize a system with two trigger mismatches (RRp2), however, it is possible to observe toehold binding and invasion processes independently (Supplementary Fig. 3C), as the mismatches prolong the time between initial binding and strand invasion to about 100 ms.

We obtained the time for complete invasion from an exponential fit to the relaxation phase of the force drop (see Fig. 2C, middle trace). For DNA, we find that strand invasion transitions triggered by an invader occur very fast and cooperatively within typically $\approx 10$–$100\ \mu s$ (mean: $42 \pm 5\ \mu s$, $N = 20$, 20 molecules, standard error of the mean (s.e.m.), blue circles in Fig. 2D). The measured transition times for DNA were very close to the response time of our instrument (gray symbols and line) given by the relaxation of the beads in water, which we measured by autocorrelation analysis[51]. We conclude that at the forces applied in our experiment, the whole strand invasion process covering 36 base pairs occurs within less than or equal to 42 microseconds. Given that backward invasion steps are highly unlikely under the high forward biasing forces (see Supplementary Methods "Strand displacement and mean first passage time for a 1D random walk" for more details), our results suggest an upper limit of 1.2 µs (42 µs/36) for a single step of invasion at forces of $\approx 10$ pN.

For RNA, the total time of the invasion process is significantly longer owed to an intermediate with a millisecond lifetime (see Fig. 2C, bottom trace, and Supplementary Fig. 4). In a buffer containing 20 mM MgCl$_2$, we find an average total invasion time of $1.39 \pm 0.08$ ms, ($N = 49$, 6 molecules, s.e.m.) (dark green circles and triangles in Fig. 2D). Note that the green triangle symbols were not measured with the fully complementary sequence but rather with a trigger strand carrying 2 proximal mismatches (RRp2). The reason for introducing these mismatches was to allow backward invasion at low loads to increase the number of data points that could be obtained with a single molecule. The duration of the invasion event will not be largely affected since it will still have to proceed through 34 of the 36 base pairs. In the absence of MgCl$_2$, the intermediate is still populated, albeit with shorter lifetimes, and the average total invasion time drops to $630 \pm 110\ \mu s$ ($N = 5$, 5 molecules, s.e.m., dark green squares in Fig. 2D). We find that the invasion times are largely independent of force (Fig. 2D and Supplementary Fig. 5), indicating that the intermediate state pausing the invasion process may be due to secondary structure formation in the trigger strand, which is not subject to mechanical force.

Since the total invasion time in RNA is dominated by the intermediate state where branch migration is halted, we tried to estimate the timescale on which branch migration proceeds by an

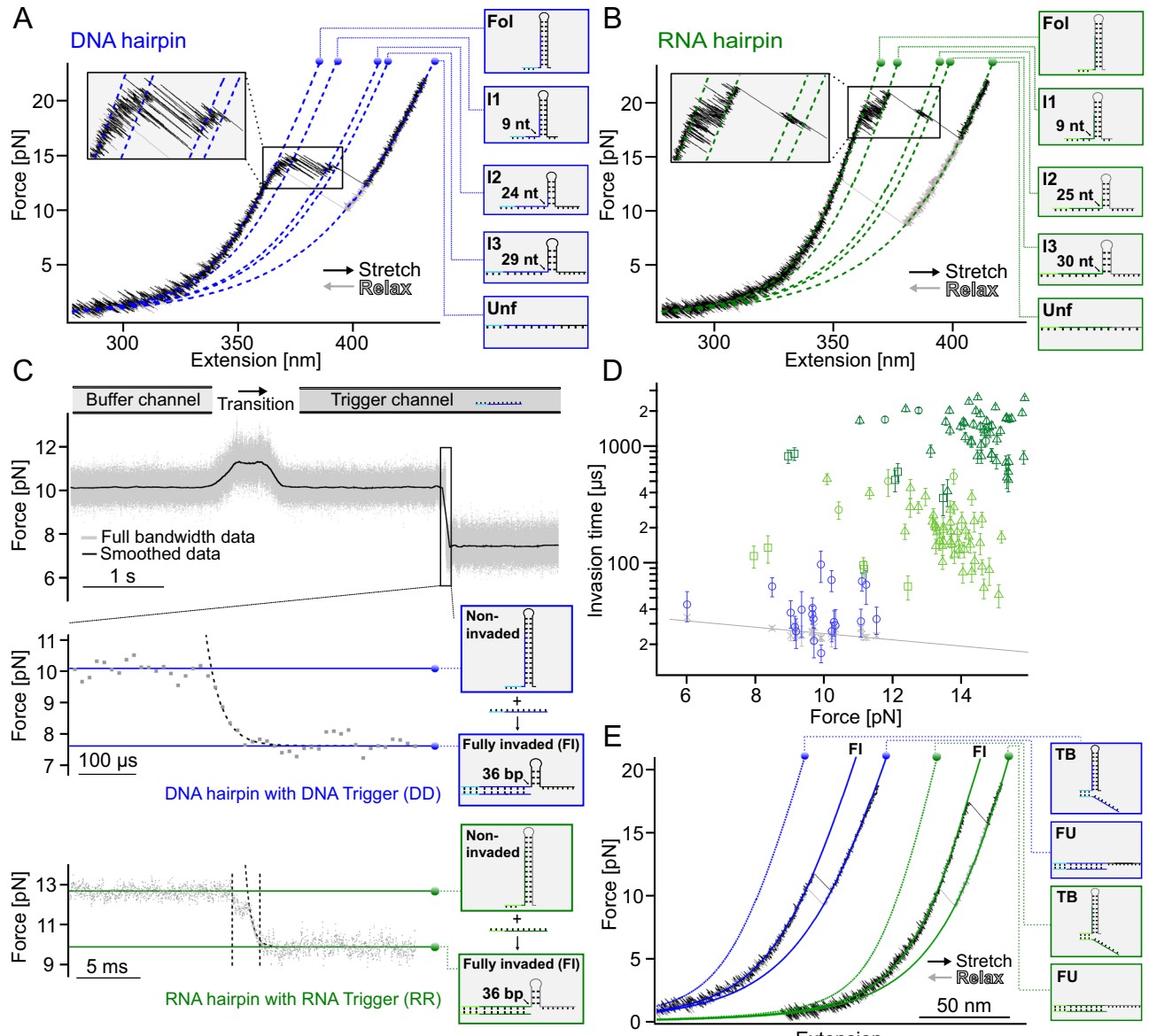

**Fig. 2 | TMSD for DNA and RNA. A** Representative force-extension curves (stretching: black, relaxation: gray) in the absence of trigger strands. The colored curves correspond to fits of a WLC model to the data for the different intermediate states. The inset shows unfolding transitions in more detail. **B** Analogous for RNA. **C** Passive mode trace of a DNA (upper trace with zoom) and RNA hairpin (lower trace) when moving from buffer to trigger channel. Binding of a trigger occurs after a lag time of ≈ 2 s, followed by toehold-mediated strand invasion, which is observed as the force drops. An exponential fit was used to determine the invasion time (black dashed line). In contrast, RNA invasion times are significantly longer (2 ms as obtained from the time between the two dotted lines), and an intermediate state is populated during the invasion process. An exponential fit to the part after the intermediate still gives longer invasion times compared to DNA (black dashed line). **D** Plots of invasion time vs. force. Gray crosses: system response time (see Methods

for details, $N = 20$). To guide the eye, we used an exponential function to fit these data points (gray line). Blue circles: DNA hairpin with DNA trigger ($N = 20$). Dark green circles: RNA hairpin with RNA trigger ($N = 3$). Dark green triangles: RNA hairpin with an RNA trigger with two proximal mismatches (see Supplementary Fig. 1 for sequence details, $N = 46$). Dark green squares: RNA hairpin with RNA trigger without magnesium ($N = 5$). Invasion times via fitting the part after the intermediate. Light green circles: RNA hairpin with RNA trigger ($N = 3$). Light green triangles: RNA hairpin with an RNA trigger with two proximal mismatches ($N = 46$). Light green squares: RNA hairpin with RNA trigger without magnesium ($N = 5$). Data are presented as mean values ± standard deviation (s.d.). **E** Representative force-extension curves after TMSD. Colored curves correspond to WLC fits: toehold bound (TB), fully invaded (FI) and fully invaded as well as fully unfolded state (FU) state. Source data are provided as a Source Data file.

exponential fit to the part of the relaxation trace following the intermediate (see Fig. 2C, bottom trace). In the presence of 20 mM MgCl₂, we find an average invasion time of $213 ± 17 \, \mu s$ ($N = 49$, 6 molecules, s.e.m.) (light green circles and triangles in Fig. 2D). In the absence of MgCl₂, the average invasion time drops to $99 ± 14 \, \mu s$, ($N = 5$, 5 molecules, s.e.m.) (light green squares in Fig. 2D). As done above for DNA, we calculated the time for a single step of RNA invasion of 5.9 μs at forces of ≈ 14 pN in MgCl₂, and 2.8 μs at forces of ≈ 10 pN in the absence of MgCl₂.

## Direct observation of repeated forward and backward invasion under force

For both DNA and RNA, the hairpin stayed permanently invaded after TMSD, and even a drastic reduction of force would not lead to a reversal of the invasion. Consequently, stretch/relax cycles in the buffer channel with the trigger-bound complex (Fig. 2E) show the molecule always in the fully invaded state (FI), indicating that backward invasion does not happen at non-zero force values. The additional unfolding transition we observe at ≈ 12 pN (DNA) and ≈ 17 pN

(RNA) merely reflects the unfolding of the remaining hairpin from the FI to the fully unfolded state (FU).

To allow observation of repeated forward/backward invasion steps close to thermodynamic equilibrium, we sought to disfavor forward invasion over backward invasion by introducing sequence mismatches into the center of the branch migration domain of the invader strand (red base in Fig. 3A)[26,52]. Such a mismatch will raise the free energy of the fully invaded (FI) state over the toehold-bound (TB) state because it possesses one complementary base pair less. Application of force will now introduce an additional intermediate state at the mismatch position (IM). The force can be chosen such that the IM state and the FI state have the same free energy, and continuous forward/backward invasion between IM and FI will be observed. In comparison to a fully complementary trigger strand (Figs. 2, 3, 1st trace), using a trigger sequence with a single $G \rightarrow T$ mismatch at position 19 of branch migration domain b' (DDc1, for sequence details, see Supplementary Fig. 1), we now observe the expected rapid forward/backward invasion equilibrium at forces around 3-4 pN (Fig. 3B, 2nd trace, forward/backward invasion marked in blue). Analysis of the involved contour length changes confirms the structural interpretation of the various states (see Supplementary Tables 3 and 4). An additional mismatch on the trigger strand introduced adjacent to position 19 (DDc2, for sequence, see Supplementary Fig. 1) shifted the force fluctuations observed from $\approx$ 3-4 pN to $\approx$ 5 pN (Fig. 3B, 3rd, trace, purple).

We also investigated the effect of an analogous mismatch ($G \rightarrow U$ mismatch at position 19) in an RNA trigger strand invading an RNA toehold hairpin (RRc1, for sequence, see Supplementary Fig. 1). The force at which the invasion starts is shifted to even higher values ($\approx$ 10 pN) and the kinetics of forward/backward invasion is slowed down so that multiple forward/backward invasion events were rarely observed at the timescale of our pulling experiments (Fig. 3B, 4th trace, pulling velocity 0.2 µm/s green). For this reason, pulling and relaxation traces for RRc1 displayed a strong hysteresis because the structure remained in the FI state until the force was reduced to values as low as 3 pN. Traces for an RNA construct (RRc2) analogous to DDc2 can be found in Supplementary Fig. 5. These traces exhibit an even larger hysteresis, and pulling forces as high as 14 pN are required to induce branch migration (Supplementary Fig. 5B).

For a quantitative assessment of the kinetics of this repeated branch migration process, we performed passive mode measurements with all three systems at different forces. Figure 3C shows sample traces at forces where IM and FI states were populated to 50 % (termed $F_{avg,1/2}$ forces in the following). The higher forces of DDc2 vs. DDc1, as well as the significantly slower kinetics in RRc1, are readily visible. Sample traces for other forces are shown in Supplementary Fig. 7.

A plot of the forward and backward invasion rates measured at different forces allows extrapolation to zero force (Fig. 3D, dashed lines). Zero force rates show that the double mismatch slows down the invasion of DDc2 by an order of magnitude as compared to DDc1 ($0.053 \pm 0.015 \, s^{-1}$ ($N = 17$, 1 molecule (shown in Fig. 3D middle), s.d. of the fitting parameter) vs. $1.57 \pm 0.12 \, s^{-1}$ ($N = 21$, 1 molecule (shown in Fig. 3D top), s.d. of the fitting parameter)). In contrast, backward invasion rates are affected less ($1500 \pm 400 \, s^{-1}$ ($N = 17$, 1 molecule (shown in Fig. 3D middle), s.d. of the fitting parameter) vs. $900 \pm 60 \, s^{-1}$ ($N = 21$, 1 molecule (shown in Fig. 3D top, s.d. of the fitting parameter)). The pronounced effect on invasion rates can be readily understood given the much higher barrier the invading strand has to overcome when two base pairs need to be broken before an invasion can move forward compared to only one. The slightly lower extrapolated rates for backward invasion in the case of DDc1 may reflect that backward invasion for the double mismatch has essentially one base pair less to compete with the invader strand and hence occurs faster. A summary of all measured and extrapolated rates can be found in Supplementary Table 5.

The extrapolated value we find for backward invasion at zero force can provide an estimate for an upper limit of the speed of branch migration per base pair. We find backward invasion rates of $\approx$ 1000/s (weighted mean: $780 \pm 30 \, s^{-1}$ (DDc1, $N = 102$, 9 molecules, s.e.m., see Supplementary Table 5) and $1480 \pm 220 \, s^{-1}$ (DDc2, $N = 60$, 6 molecules, s.e.m., see Supplementary Tab. 5)) for branch migration across 17 (DDc1) or 16 (DDc2) bases and, hence, an upper limit of $\tau_{0,D} \leq 59 \, \mu s$ ($0.5 \times (1/(780 \, s^{-1} \times 17) + 1/(1480 \, s^{-1} \times 16)) = 58.8 \, \mu s$). Note that for calculating the upper limit and since the process occurs under a strong biasing force, we assume linear scaling with the number of bases covered (see discussion in the Supplementary Methods "Strand displacement and mean first passage time for a 1D random walk").

Comparison between DDc1 and RRc1 (Fig. 3D top vs. bottom) shows that both values for forward invasion extrapolated to zero force ($1.57 \pm 0.12 \, s^{-1}$ vs. $0.009 \pm 0.003 \, s^{-1}$), as well as backward invasion ($900 \pm 60 \, s^{-1}$ vs. $230 \pm 90 \, s^{-1}$) are slower for the RNA construct. The lower value we find by extrapolating the backward invasion branch directly shows that branch migration in RNA is considerably slower than in DNA. The same estimate as that used in the previous paragraph yields a value for RNA branch migration that is slower by a factor of 4.4 ($\tau_{0,R} \leq 260 \, \mu s$). For RRc2, conformational transitions recorded in passive mode experiments were too slow to allow an accurate estimate of the corresponding kinetic rates (Supplementary Fig. 5C).

## Simulation of force−extension curves and energy landscapes

To aid the interpretation of our experimental results on force-induced TMSD and help understand the impact of mismatches in the invader sequence, we studied the TMSD process using oxDNA simulations. OxDNA[44,45] is a coarse-grained DNA model that represents each nucleotide as a single rigid body with interactions parameterized to reproduce structural, thermodynamic, and mechanical properties of single-stranded and double-stranded DNA. The model has been extensively employed to simulate strand displacement processes and has been shown to be in good agreement with experiments[26,53,54] and has also been shown to accurately capture the mechanical response of DNA to tension[55,56]. We also employed the RNA version of the model, oxRNA[57], to study the RNA TMSD under force.

We began by creating DNA constructs composed of DNA handles, adapters, DNA hairpins, and trigger strands using the oxView tool[58]. We then performed molecular dynamics (MD) simulations in the absence of trigger DNA, in which we applied tension to both ends of the DNA handle and pulled on the toehold hairpin with a constant velocity of 0.14 mm/s (Fig. 4A, left). The pulling velocity is larger than the experimental one, as the experimental pulling rates are hard to achieve in MD simulations. The calculated force-extension curve showed hairpin unfolding transitions around 22 pN. This value is higher than the forces observed in experimental results (Fig. 2A), likely due to the faster pulling rate in the simulation. Snapshots of the simulated DNA molecule visualize the state of the hairpin at different forces attained during the stretching process.

When we start the pulling simulation with a fully complementary trigger strand (DD) bound to the toehold, strand invasion by the trigger promotes the unfolding process, and invasion is finished before reaching 10 pN (Fig. 4A, right). The short unfolding transition at around 20 pN corresponds to the force-induced unfolding of the remaining stem (see overlay in Supplementary Fig. 12A for comparison between DD and without trigger).

Furthermore, during the branch migration process on both our DNA and RNA model, we observed the formation of secondary structures on the trigger strand (Fig. 4B and Supplementary Fig. 12B, C). The secondary structures we found for RNA triggers were much more stable than those for DNA. We hypothesized that they could explain the slower branch migration rate and intermediate state we observed experimentally during RNA/RNA as compared to DNA/DNA strand displacement (Fig. 3D and Supplementary Figs. 3, 4, and 5).

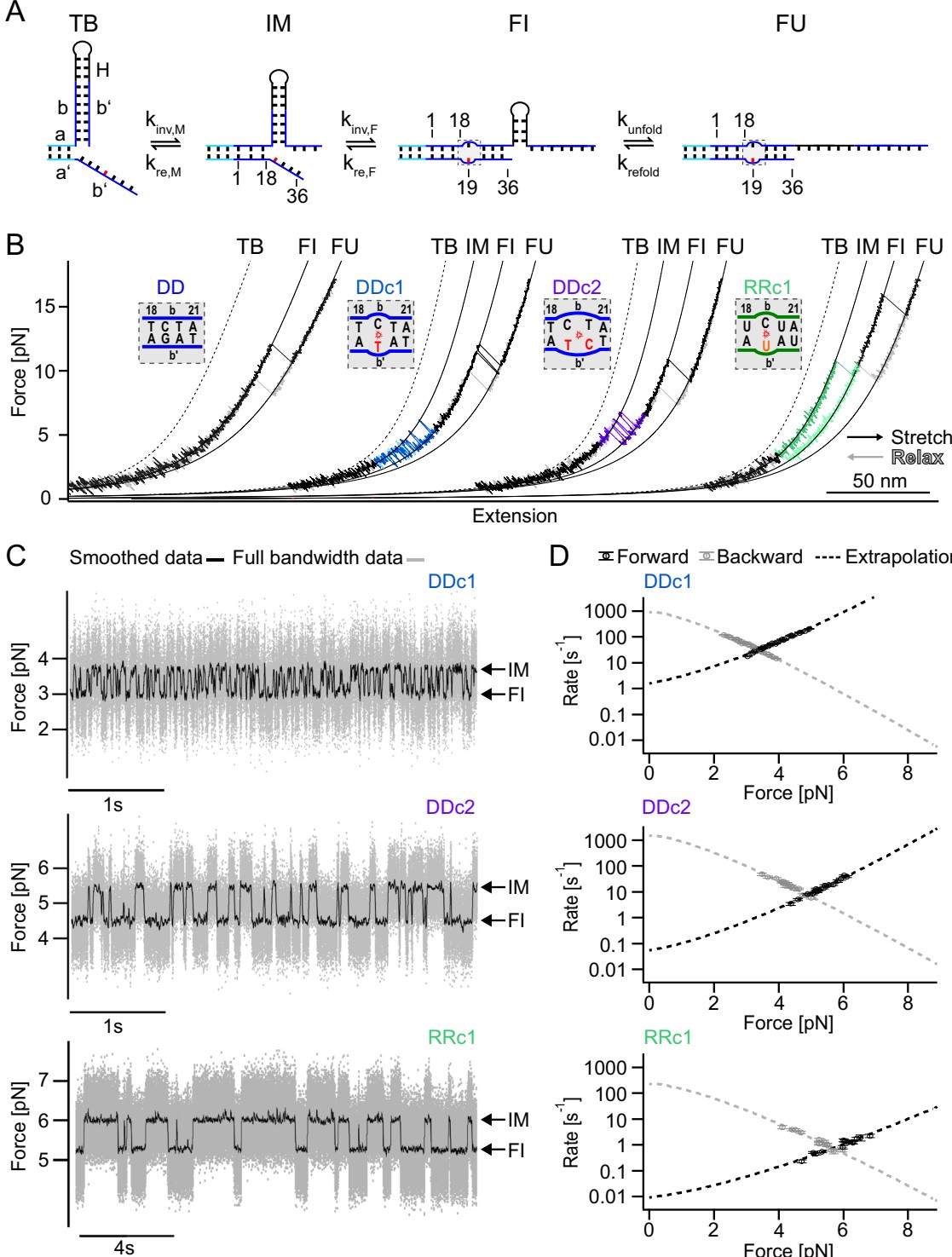

**Fig. 3 | Direct observation of repeated forward/backward invasion by introducing trigger mismatches in SMFS experiments. A** Conformational states during invasion of a mismatched trigger strand into a toehold hairpin: Toehold-bound (TB); invaded until the mismatch position (IM); fully invaded (FI); fully unfolded (FU). The mismatch is highlighted in red (DNA) and orange (RNA). **B** Force-extension traces of a DNA toehold hairpin with a fully matched DNA trigger (DD, first trace), DNA trigger strands with one or two mismatches (DDc1, second trace, and DDc2, third trace, respectively), and an RNA toehold hairpin with an RNA trigger strand with one mismatch (RRc1, fourth trace). For better visualization, each trace is horizontally shifted with a constant offset. The insets show the mismatched positions and sequence details. **C** Force-versus-time traces recorded at $F_{avg,1/2}$ for the mismatched trigger strands DDc1, DDc2, RRc1, showing several forward/backward invasion transitions between the IM and FI states due to strand displacement. **D** Force dependence of forward and backward invasion transition rates between the IM and FI states. The dotted lines represent extrapolations of the data based on a model described in the SI ("Model for transition rate-extrapolation"). Data points of one sample molecule are shown. Weighted averages of the fitted parameters of all molecules are summarized in Supplementary Table 5. Source data are provided as a Source Data file.

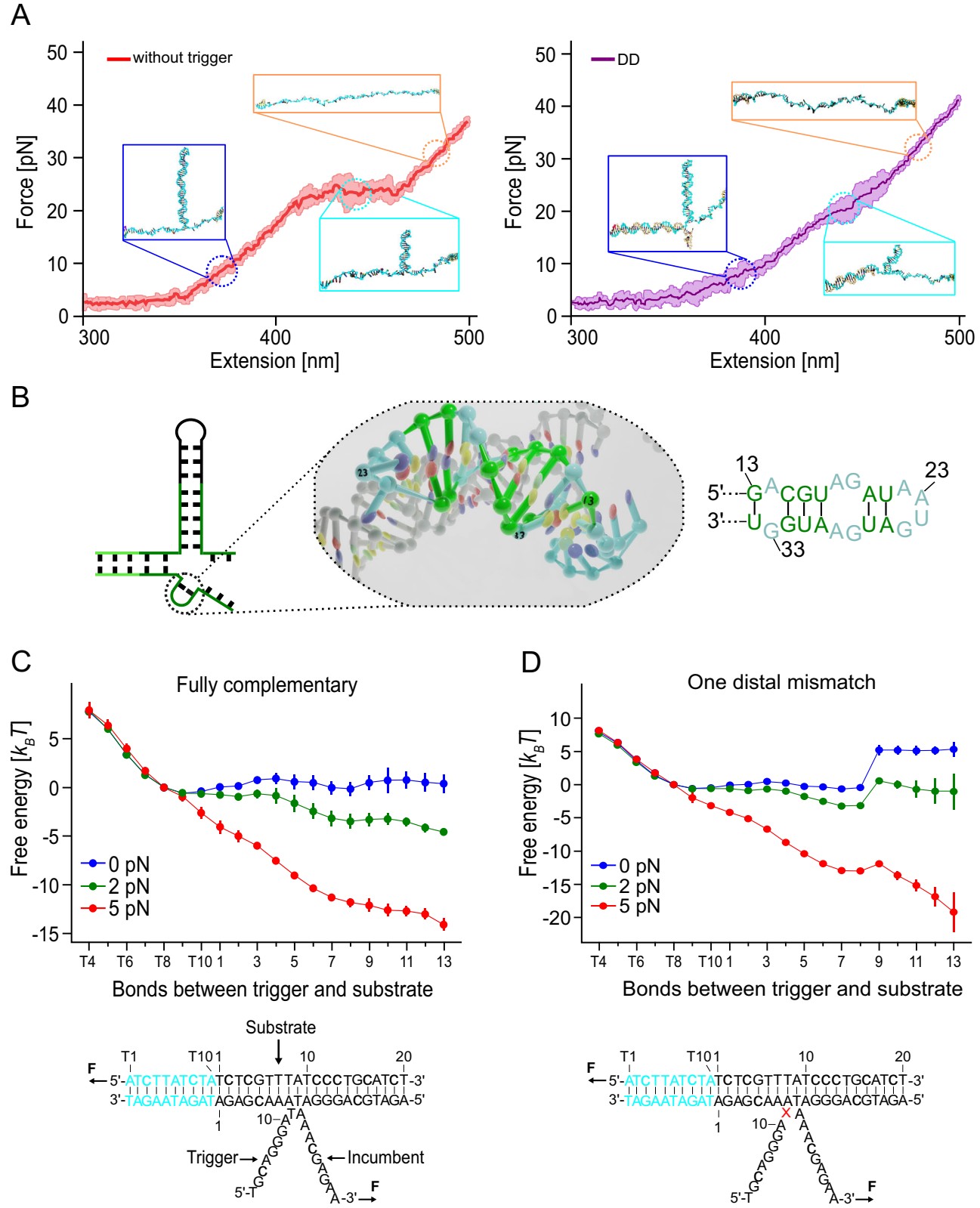

We next performed molecular dynamics simulations of the full system where we applied constant force (ranging from 2 pN to 10 pN) on the DNA hairpin (see Supplementary Methods "Simulation protocols" for more details). Simulations with different trigger strand variants demonstrated that consistent with our experiments, increasing pulling forces generally accelerated the overall kinetics of the branch migration process and biased it toward full displacement

(Supplementary Fig. 12D, E, and F). For example, when a pulling force over 5 pN was applied, the single and double central mismatches (DDc1 and DDc2) could be overcome more easily than at 2 pN. These findings offer further evidence that the impact of mismatches can be fine-tuned by varying the applied pulling force.

The single-nucleotide resolution of the oxDNA simulation can further provide a better qualitative understanding of the free-energy

**Fig. 4 | Simulation of force-extension traces of the toehold hairpin and free-energy landscape of TMSD with mismatched sequences using oxDNA.**
**A** Averaged force-extension curve was obtained from pulling the DNA construct at a constant speed of 0.14 mm/s, comparing the toehold hairpin (red) and with a fully complementary trigger strand (violet). The simulation has 7 replicas. The curves represent the mean values, and the outline of the curves represents ± 1 s.d. Snapshots of the DNA constructs are shown to depict the intermediate states observed during the unfolding process. **B** The most formed base pairs of RNA trigger strands during the SD are shown on both the predicted secondary structure and 3D

structure (marked in green). **C, D** Free-energy landscapes for different trigger strands: fully complementary (**C**), one distal mismatch (**D**). **C**: $N = 10$, **D**: $N = 10$. Data are presented as mean values ± standard deviation (s.d.). The DNA constructs' sequences are depicted below, with the mismatches highlighted in red. The coordinates represent the specific pairing interactions between the target sequence and the trigger sequences in each state of the system. The free-energy landscapes are shown for various force conditions, distinguished by different colors. An additional adenine was added to the 3' end of the incumbent strand, where the force is directly applied to. Source data are provided as a Source Data file.

landscape of the system when subjected to an externally applied force. The free-energy sampling simulations of the full system are, however, very compute-intensive, and we hence studied a simplified system where we reduced the length of both the toehold hairpin and trigger strand. We focused on a three-stranded system with a total of only 30 bp (Fig. 4C, D), and we applied constant forces ranging from 0 to 5 pN to the strands. We obtained the free-energy landscapes from simulations as a function of the number of base pairs formed by the target strand with the trigger and the incumbent for different force biases (Fig. 4C, D) using umbrella sampling[53] (see Supplementary Methods "Simulation protocols" for more details).

From the free-energy landscapes, we observe that as the force increases, the states with more base pairs formed between the invader and the substrate become more favorable. A 2D free-energy landscape as a function of the number of base pairs formed between invader and substrate and incumbent and substrate, respectively, is shown in Supplementary Fig. 11A, B, and C. The larger the applied force, the more favored the states with the trigger bound to the incumbent, as seen in the 2D projections of the free-energy landscapes. We further note that the free-energy landscape (Fig. 4C) shows a small local minimum at around 8 base pairs formed with the trigger strand that coincide with longer waiting times in the MD simulations. These likely originate from a sequence-dependent effect of poly-A stacked regions in the trigger strand. The position of the minima coincides with the increased state occupancy observed in the kinetic simulation (see Supplementary Fig. 12D and E). However, these local minima in the free-energy landscape are not expected to have a measurable effect in the experiment.

We further investigated the effect of mismatches on the free-energy landscape for different applied forces. We then simulated a proximal mismatch, similar to the experimentally studied mismatch shown in Supplementary Figs. 3 and 6. The mismatch between the trigger and incumbent strand creates a barrier to the displacement (Supplementary Fig. 11D), and two mismatches (Supplementary Fig. 11E) further increase the barrier ($\approx 5\, k_BT$ (1 mismatch) vs. $\approx 12\, k_BT$ (2 mismatches)). In the case of one proximal mismatch and 2 pN force (Supplementary Fig. 11D, red trace), the non-invaded state (T10) and "fully" invaded state (13) have almost the same free energy. In the case of the distal mismatch (Fig. 4D), we observe a barrier of $\approx 5\, k_BT$ at the position of the mismatch, which is reduced by applying increasing pulling force.

**Force-induced TMSD in a DNA/RNA hybrid**
Compared to B-form DNA duplexes, double-stranded RNA and RNA-DNA hybrids adopt an A-form helix conformation[59]. However, double-stranded RNA is significantly more stable than RNA-DNA hybrids (Fig. 5A middle vs. right), which makes invasion into an RNA stem by a DNA invader free-energetically unfavorable[60–62]. In the absence of force (black free-energy landscape Fig. 5B), the strand invasion process of DNA replacing RNA in an RNA duplex is free-energetically unfavorable and unlikely to happen spontaneously. We surmised that in SMFS experiments, tilting the free-energy landscape by an applied pulling force would allow us to equilibrate forward and backward invasion processes and thus enable observation of strand displacement by a DNA trigger (Fig. 5B gray free-energy landscape).

Figure 5C shows a passive mode experiment where a fully complementary DNA trigger invades the RNA hairpin (RD) at an average force $F_{avg,1/2} = 10.6$ pN. In contrast to the DD and RR constructs measured in Fig. 2, we can observe repeated forward/backward invasion transitions occurring close to equilibrium. These transitions occur via pronounced intermediates TB, RD1, RD2, RD3 and FI. Similarly, pulling/relaxation cycles show the same intermediates populated close to equilibrium (Supplementary Fig. 8). It is important to note that all intermediates occur at significantly lower forces compared to the intermediates populated during unzipping of the RNA hairpin in the absence of a trigger (compare Fig. 2B). Moreover, those intermediates also occur at different positions with the possible exception of RD1 (8 vs. 9 bp invaded). Our finding of several intermediates for the RD construct strongly indicates that sequence effects play an essential role in invasion. We computed the relative free-energy differences between each pair of intermediate states by Boltzmann inversion of the population probabilities, which confirmed that for an applied average force of 10.6 pN, all states were roughly at the same free energy (Fig. 5D, gray). Including the measured transition rates between the various states allowed us to extract also transition state energies and construct a schematic free-energy landscape assuming an Arrhenius pre-factor of $3 \times 10^6\, s^{-1}$ [63] and transition state positions in the middle between the states (gray free-energy landscape in Fig. 5D). Force dependent rates and extrapolations to zero load are shown in Supplementary Fig. 8. Transformation to zero load yields the black free-energy landscape in Fig. 5D. The difference in free energy between TB and FI in the black free-energy landscape of $\approx 40\, k_BT$ is consistent with the expected free energy required to fully break an RNA duplex until position 36 with subsequent formation of an RNA/DNA hybrid (cf. Supplementary Tables 2 and 7). Under a load of $F_{avg,1/2}$, all lifetimes between transitions are on the order of milliseconds; however, when reducing the load, backward invasion rates will quickly win over forward invasion rates, and the equilibrium will strongly shift towards the TB state.

## Discussion

Strand displacement processes have been previously studied predominantly using bulk kinetic measurements[2,19–21], with the exception of only a few single-molecule studies conducted with single-molecule FRET techniques[64] or magnetic tweezers[41]. Adding to these experiments, optical tweezers-based single-molecule force spectroscopy, such as applied in the present study, provides a detailed glimpse into the process and, together with the OxDNA simulations, allows close-to-base-pair resolution. While the results of our single-molecule TMSD experiments are in line with previous bulk studies, they provide detailed insights into sequence-specific features that affect the TMSD process. In the following discussion, we highlight several of the most remarkable findings that were extracted from the SMFS measurements and OxDNA simulations.

**Time-scales of TMSD**
Compared to other recent SMFS studies[41,65,66], our experimental setup offers nearly an order of magnitude faster response time, allowing us to capture branch migration rates closer to the true kinetics. Direct observation of strand invasion in passive mode experiments at a force

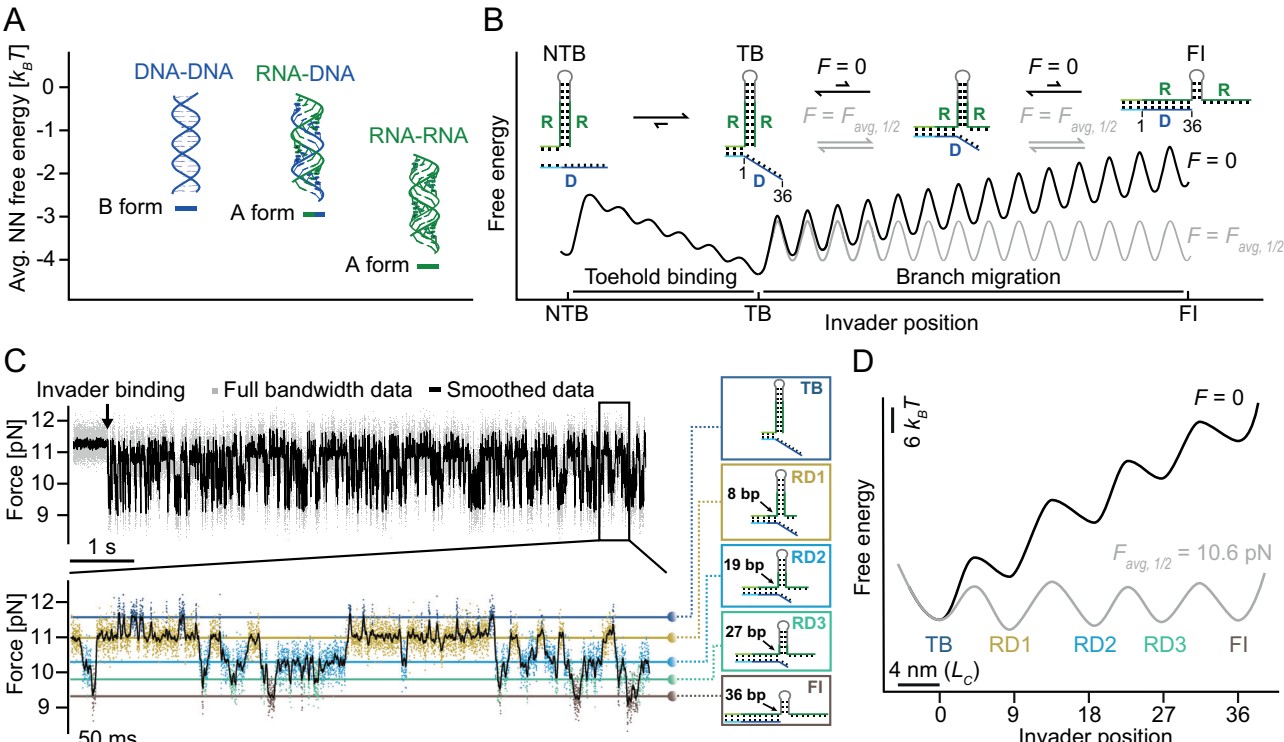

**Fig. 5 | Force-induced DNA-RNA hybrid TMSD and free-energy landscapes at $F_{avg,\frac{1}{2}}$ and zero load. A** Average nearest neighbor free energy per base pair for DNA, RNA, and DNA-RNA hybrid. **B** Schematic of force-induced DNA invading RNA process. DNA trigger invasion to RNA hairpin is unfavorable due to the free-energy difference between RNA-RNA stem and DNA-RNA strand. The free-energy landscape (black) can be tilted by force, resulting in a flat free-energy landscape (gray) with equal forward/backward invasion rates. **C** Force versus time trace of DNA-RNA hybrid TMSD. Each intermediate state during the branch migration process is distinguished by color. TB: toehold bound (dark blue) DR1, 2, 3: intermediates 1 (yellow) 2 (light blue), 3 (green), FI: fully invaded (brown). **D** Extracted free-energy profiles: almost flat landscape at 10.6 pN and uphill landscape at zero load. Source data are provided as a Source Data file.

bias ($F \approx 10$ pN) resulted in strand displacement times for a 36 bp long domain of $\approx 42\,\mu s$ for fully complementary DNA triggers. It is important to note that this number is rather an upper bound since our instrumental limitation is in a similar range ($\approx 25\,\mu s$). $42\,\mu s$ for full completion of the invasion process appears surprisingly fast compared to estimates based on step times derived from bulk FRET experiments ($28\,\mu s$ per single step)[54]. At a $28\,\mu s$ step time, the full invasion process covering 36 base pairs would take $\frac{1}{2} \times (36 \times 37) \times 28\,\mu s \approx 19$ ms (compare Supplementary Methods "Strand displacement and mean first passage time for a 1D random walk"), almost 3 orders of magnitude slower than the number we find. Two major effects can account for this apparent discrepancy: first, at zero force, strand displacement is a diffusive process and scales with the square of the number of steps, while at $\approx 10$ pN of force, strand displacement will be strongly biased and hence, scale linearly with the number of steps (for details see Supplementary Methods "Strand displacement and mean first passage time for a 1D random walk"). This conclusion is supported by our constant force simulations, as backward steps at 10 pN force were almost never observed (Supplementary Table 8 and Supplementary Fig. 12F). Moreover, the force will affect the rate of branch migration by lowering the barrier of each individual step and hence speed up the displacement process (Supplementary Equation 20). Our simulations show a small increase in the forward stepping rate and a huge decrease in the backward stepping rate with increasing force (Supplementary Table 8 and Supplementary Fig. 12F). Another cause for faster displacement under force could be a change in the displacement mechanism where force could break more than one base pair at a time, and the process proceeds in larger effective step sizes. The kinetic simulations, however, still support an invasion process where for every broken base pair between the incumbent and the substrate strands, a new base pair is immediately created between the substrate and invader, even at 10 pN. Our findings of significantly faster branch migration under load could be important in a context where DNA machinery applies forces to DNA while another strand invades[30,67,68]. Even though the mechanisms for TMSD change upon application of force from diffusive to directed, our data provide a direct comparison of the single step time between DNA and RNA (Fig. 2D). For DNA, at a force of 10 pN, we find a single step time of $1.2\,\mu s$ (full invasion into 36 bp to be completed in $42\,\mu s$), and the single step time for RNA is $5.9\,\mu s$ at 14 pN slower by a factor of 4.9.

Experiments conducted with mismatched triggers allowed us to directly observe transitions between the IM and FI state, which were used to estimate an upper bound for the duration of strand displacement processes at zero load (Fig. 3). As detailed in the results section, we arrive at an upper limit of $\tau_{0,D} = 59\,\mu s$ per single branch migration step for DNA, which is close to faster estimates from bulk experiments obtained at zero load[20,21,26,53,54]. Their single-step times were only inferred from bulk measurement rates, however, potentially convoluting multiple effects in the fitting, including association/dissociation rates and fluorescent reporter kinetics. Regarding single step times in RNA, one study reported similar values as for DNA[27]. In our work, however, we find an upper bound for the single step time of RNA branch migration of $\tau_{0,R} = 260\,\mu s$, a factor of 4.4 slower than DNA. The slower step times we find for RNA compared to DNA are consistent with a free-energy landscape model where the branch migration step involves larger free-energy barriers than in the case of DNA, possibly due to the more stable base pairing in the A-form helices of RNA. Interestingly, the factor by which single step times differ between DNA and RNA is very similar in this estimate to the one in the previous paragraph.

A major difference between RR strand invasion as compared to DD is the occurrence of an intermediate state populated during branch migration (see Supplementary Fig. 4 sample traces for RR, Supplementary Figs. 5 and 6). Our analysis of the invaded nucleotide distribution for individual strand invasion events in Supplementary Fig. 6D shows two intermediate candidates, one at ≈ 8 bp and another one at 15 bp. Possible explanations include a sequence-related free-energy minimum at the position where we find the intermediates or structure formation in the invading strand, which is not under load in our experiment. We have several pieces of evidence that, indeed, structure formation in the trigger strand leads to the intermediate states. First, the force dependence of the duration of the intermediates is very weak (see Fig. 2D, invasion times of dark green symbols are largely independent of force). This supports the idea that structure forms in the part that is not under load, i.e., the trigger strand. Additional support for structure formation in the invading strand comes from the analysis of the traces for the RRc2 construct (Supplementary Fig. 5). This construct is designed to form an intermediate IM at the position of its 2 mismatches. However, the traces show that backward invasion can consistently go further back to another intermediate BI at 14 bp, close to the intermediate at 15 bp from RRp2. A natural explanation for this behavior is that whenever structure forms in the invader, base-pairing of the involved bases with the hairpin is blocked, thus allowing the backward invasion to proceed further back than the mere mismatch would allow. Supporting the idea of secondary structure formation, our MD simulations in oxRNA reveal a stable secondary structure featuring six base pairs on the RNA trigger strand (Fig. 4B). As in our experiments, this stable secondary structure acted as an impediment. While in our experiments, strand displacement was delayed, in the simulations secondary structure effectively blocks the entire strand displacement process. Our results of finding transient secondary structure formation in the trigger strand are somewhat surprising since we optimized the trigger strand specifically against such secondary structure formation when designing the construct using NUPACK. Even though this secondary structure is only populated on a millisecond timescale, it may still be important to take into account in future designs of RNA riboregulators. Note that the intermediates also occur in the absence of magnesium, albeit with shorter lifetimes (630 μs vs. 1.4 ms) (Fig. 2D and sample traces Supplementary Fig. 4).

## Times for RNA/DNA hybrid

Owing to the unfavorable free-energy balance in favor of RNA double strands, DNA strand invasion into a long (36 bp) fully complementary sequence is completed fast (average times of 69 ms) only under mechanical forces of 10.6 pN (Fig. 5). In the absence of force, it will not occur spontaneously (once every $5.1 \times 10^{13}$ s or 1.6 million years as calculated using Supplementary Equation 16). This time will be shorter if shorter sequences are used. For example, invasion into the first 8 bps from TB to RD1 will only take 0.74 s even at zero force (Supplementary Fig. 8). Hence, adapting the length of the branch migration domain offers a possibility to tune the desired kinetics. DNA as an invader has the advantage of less likely secondary structure formation as compared to an RNA invader strand. Note that introducing interior bulges into the stem of an RNA toehold hairpin offers another possibility of tuning invasion kinetics by shifting the free-energy balance such that a fully complementary DNA invader can rapidly invade an RNA toehold hairpin even at zero load. Using the original toehold switch structure[15], we show that DNA invasion into RNA can indeed shift the equilibrium toward the open state of the toehold hairpin (Supplementary Fig. 9).

In contrast to the pure DD and RR systems investigated in this study, the RD hybrid exhibits a pronounced sequence effect, leading to three distinct intermediates. We can rule out secondary structure formation as a cause since the invader is DNA and neither in our DD experiments nor in the simulation we observed any significant formation of secondary structure. We attempted a simple calculation of these sequence effects by the nearest neighbor model shown in Supplementary Fig. 8F at zero load and Supplementary Fig. 8G under load. Indeed, the hybrid shows sequence modulation of the free-energy landscape, which is completely absent in the nearest neighbor model for DD and RR constructs. However, the calculated effect is still too small to explain the significant barrier heights as well as free-energy differences we find experimentally (Fig. 5D). More elaborate models will be necessary to understand the sequence dependence we find quantitatively.

A coarse estimate of an upper bound for the single step time for backward invasion by the RNA strand from the extrapolation to zero load of the upper branch in the rate plots of Supplementary Fig. 8B–E leads to $\tau_{0,RD}$= 4 μs. This estimate is even faster than the upper bound we find for DNA/DNA branch migration using the same method. Note that much higher forces had to be applied to allow strand displacement to happen in equilibrium for the RD hybrid as compared to the DNA/DNA and RNA/RNA constructs, and the estimate will be less reliable since it involves extrapolation across a much larger force range. Moreover, the probability of faithful base pair by base pair invasion might be decreased, and a process involving steps where larger units break together might be favored.

## Effect of mismatches on branch migration kinetics

Introducing mismatches is a widely used method to tune the branch migration kinetics in DNA nanotechnology[26,52]. We utilized sequence mismatches on the trigger strands to allow repeated observation of forward/backward invasion under load. From the kinetics measured under load (Fig. 3D), we could then extrapolate back to the kinetics at zero load. While in the fully complementary constructs, forward invasion happened quasi instantaneously after the toehold had bound, rates for forward invasion at zero load were slowed down considerably for both DDc1, DDc2 as well as RRc1 (2 s$^{-1}$ at zero load for DDc1, 0.05 s$^{-1}$ for DDc2 and 0.009 s$^{-1}$ for RRc1, see Supplementary Table 5). The rate plots in Fig. 3D directly provide a measure for the change in free energy the mismatches induce compared to the fully complementary sequences. The calculated free-energy differences are summarized in Supplementary Table 5. The values are within 18 % of the prediction by the nearest neighbor model NUPACK for the DNA mismatches and 27 % for the RNA mismatch. The systematic deviation of our experimental numbers to lower values as compared to the prediction is likely due to systematic errors in our force calibration (see paragraph after Supplementary Table 2 for more details).

In previous studies, it was shown that also the position of the mismatch (proximal, central, distal) plays a role in the displacement kinetics[26,54,69]. There are two different reasons why this was observed in bulk measurements. First, a distal mismatch was shown to affect the displacement rate less drastically since the last few steps of branch migration are affected by the spontaneous dissociation of the incumbent[54]. Second, a proximal mismatch was shown to have a stronger effect for a short (not saturated) toehold, as a waiting time at the mismatch position would increase the probability of dissociation of the invader strand and would affect the effective second-order rate of binding to the toehold. In case of a mismatch further downstream, the complex of invader and target would be more stable (containing more base pairs); hence, dissociation of the invader would be less likely. Our design is independent of these effects and allows the determination of position-independent displacement kinetics for two reasons. First, with our toehold hairpin design and additional remaining stem, even after the full invasion, spontaneous dissociation of the incumbent is impossible. Second, we use a toehold length of 14 nt (the same as that used for a toehold switch in ref. 15), which allows prolonged binding of the invader to the toehold over the time of our experiment. In addition, even if the toehold length was shorter and detached during the measurement, our setup could detect these events as toehold binding/unbinding can be observed separately, at

least for a system like RRp2 (see Supplementary Fig. 3). In this case, the toehold binding is characterized by an increase in force since the stretched double-stranded toehold-invader complex has a shorter extension than the stretched single-stranded toehold. The invasion is delayed because of the proximal mismatches directly after the toehold binding domain ($\tau_{TB}$ in Supplementary Fig. 3C). After $\approx 100$ ms, the invasion proceeds, marked by a drop in force as the invader opens the hairpin.

### Significance for nucleic acid nanotechnology and synthetic biology

Our results are significant for the further development of dynamic nucleic acid nanotechnology and applications in synthetic biology in many respects. Using bulk fluorescence studies, researchers in DNA nanotechnology had previously studied the kinetics of TMSD processes in great detail. These studies extracted effective branch migration rates from bulk kinetic data by dissecting the contributions of the initial binding to the toehold and the strand displacement process itself. Bulk studies consistently showed that the kinetics are strongly affected by mismatches in the branch migration domain, both on the incumbent and the invader. Notably, such differences have already been exploited for the detection of single nucleotide polymorphisms in a wide range of DNA sensors[70] or even for RNA sensors in vivo (single-nucleotide-specific programmable riboregulators, SNIPRs)[71]. Our single-molecule studies confirm the prominent role of mismatches and, moreover, provide a direct observation of strand displacement in individual molecules under load. These experiments also suggest that the process is remarkably variable along the branch migration domain, pointing toward a strong sequence dependence that is not yet completely understood or characterized.

In particular, applications of TMSD for the realization of molecular circuitry inside living cells will require the use of RNA molecules either as inputs or as substrates for strand displacement processes, or both, resulting in situations where RNA invaders have to invade RNA or DNA duplexes, or DNA has to invade RNA duplexes or RNA-DNA hybrids. In this context, our study demonstrates that RNA invasion into RNA is considerably slower than DNA invasion into DNA duplexes. Formation of intermediate states owed to secondary structure formation may further slow branch migration in RNA invading RNA. Moreover, DNA invasion into RNA duplexes can be promoted by force.

Strand displacement processes also play a role in biological and synthetic biological contexts, e.g., in the invasion of CRISPR-guide RNA complexes into DNA duplexes the switching of riboregulators or conditional guide RNAs. While we have a general understanding of the underlying processes, it has often been difficult to interpret the widely differing efficiencies of specific sequences that superficially look very similar. This has recently led to the wide application of machine-learning approaches to support the design of functional RNA molecules[42,43,72]. Our biophysical insights may prove useful in the interpretation of the predictions made by such systems and potentially result in a set of refined but understandable design rules. Ideally, the deliberate choice of specific branch migration sequences, including the introduction of mismatches and bulges, can be used to fine-tune the performance of gene regulatory switches for specific applications.

Finally, we would like to remark that pico-Newton forces are prevalent in the biological context, under which strand displacement processes operate. RNA and DNA polymerases have been shown to be strong molecular motors that can work against forces as high as 15-34 pN[73–75]. We found that the application of lower forces already leads to a strong speed-up of strand displacement processes, suggesting that TMSD in cells might operate with considerably different kinetics than typically observed in the test tube. For instance, the forces generated by RNA polymerases during transcription of a riboregulator or a ribosome plowing through the secondary structure in an RNA molecule might either prevent or support strand invasion by a trans-acting RNA effector.

To conclude, we aimed to measure the kinetics of hairpin TMSD using a microfluidic-based single-molecule assay. Our inspiration for this approach came from the riboregulator toehold switches[15], where we designed a toehold hairpin structure and applied force on both sides of the molecule without dissociating the incumbent strand as in 3-way branch migration. This enabled us to measure the forward and backward invasion process of branch migration on a single-molecule level for many repeats, thus increasing statistical significance and allowing detailed insights into the fundamental process.

Our measurements provide a high level of temporal resolution of the dynamic process of TMSD and, together with our coarse-grained simulations, allow a detailed understanding on a base-pair level. Our findings of the dynamic process of branch migration under force, which is similar to many conditions when RNAs interact with proteins, particularly in cellular environments, can have significant implications for the design and fine-tuning of dynamically functional RNA riboregulators, as well as CRISPR-based genome editing and gene therapy.

The kinetics of the branch migration process are influenced by the force applied to the DNA or RNA strand, a condition similar to that of translation or RNA chaperone binding processes. Our study provides insights into the kinetics and thermodynamics of the strand displacement process and sheds light on the factors that affect the branch migration rate. The information gained from this study will be helpful in designing and optimizing DNA/RNA-based molecular machines and devices for various applications in biosensing, diagnostics, and therapeutics.

## Methods

### Preparation of toehold hairpin constructs

Toehold hairpins and trigger RNA molecules were produced by in vitro transcription from plasmids (for more details on the cloning process see Supplementary Methods: "Culture media", "Plasmid construction and cloning process" and "Cell culture") followed by gel purification.

**DNA handle preparation and In vitro transcription.** We PCR-amplified the DNA handle strands (545 bp) from Lambda phage DNA using modified primers (see Supplementary Tables 9 and 10). Specifically, the forward primer was labeled with two dT-Biotin or dT-Digoxigenin molecules at the 5′ end, while the reverse primer included a stable abasic-site to preserve a single-stranded overhang for binding to the target molecule.

All in vitro gene transcription experiments with toehold hairpin and trigger RNAs were performed using a homemade in vitro transcription mix, including a homemade T7 RNA polymerase. The T7 RNA polymerase with a 6xHis tag was expressed in *E. coli* BL21 DE3, followed by cell lysis using lysis buffer (1 mM Benzamidine, 1 mM PMSF, 1:2000 (1 mU) dilution Turbo DNase from Ambion, 1 mg ml⁻¹ Lysozyme of chicken egg white) and sonication, followed by purification using an ÄKTA pure Chromatography System. Next to the T7 RNAP, the TX mix contained transcription buffer (50 mM HEPES, 22 mM $MgCl_2$, 100 mM KCl, pH 7.8) and Murine RNase inhibitor (NEB) in a 20 µl or 100 µl reaction. Linear transcription templates for toehold hairpins and trigger RNAs were first amplified using PCR and purified using a Monarch® PCR Cleanup Kit (NEB). The concentration and quality of purified DNA templates were quantified via their 260/280 and 260/230 ratios using a Nanodrop 8000 spectrophotometer (Thermo Fisher). The molar concentration of each DNA template was calculated via:

$$\frac{\text{Concentration}\left(\text{ng µl}^{-1}\right) \cdot 10^6}{\text{Molecular weight}\left(\text{g mol}^{-1}\right)} = \text{Concentration(nM)} \quad (1)$$

**Table 1 | Hybridization reaction conditions**

| STEP | TEMP | TIME |
|---|---|---|
| **First hybridization** | | |
| Initial Denaturation | 70 °C | 1 min |
| Folding | 68 °C | 30 s |
| | 65 °C | 20 min (30 cycles) |
| Final | 25 °C | 5 min |
| Hold | 4–10 °C | ∞ |
| **Second hybridization** | | |
| Initial Denaturation | 68 °C | 1 min |
| Folding | 68 °C | 30 s |
| | 63 °C | 20 min (30 cycles) |
| Final | 25 °C | 5 min |
| Hold | 4–10 °C | ∞ |

In the first hybridization reaction, the toehold hairpin is annealed to a DNA handle and the adapter strand to another DNA handle. In the second hybridization reaction, the products of the first hybridization reaction are annealed together.

**Agarose gel electrophoresis and Urea-PAGE purification.** The DNA handles used for attaching the target molecule and silica beads (0.5 kb) were initially PCR amplified (primer sequence see Supplementary Table 9) and purified through agarose gel (2% wt, from CARL ROTH). However, we encountered a false priming issue that resulted in an additional 200 bp junk strand on the PCR product, which could significantly impact the subsequent folding process with the target molecule. The target bands were cut and purified using a Gel Purification Kit from QIAGEN.

After in vitro transcription, the toehold hairpin and trigger RNAs were initially digested for 30 min using DNase I (from NEB) to remove the original DNA template. Next, we added 0.5 M EDTA to chelate the remaining $Mg^{2+}$ from the samples and denatured them at 65 °C. After denaturation, the RNA samples were purified using a 10% Urea-PAGE gel (consisting of Urea 4.8 g, 40% Acryl (29:1) 2.5 ml, 30% APS 50 μL, TEMED 10 μL, and 10xTBE 1 mL, all from Carl Roth). The gel electrophoresis was performed using the Owl™ gel system. Following gel electrophoresis, the target bands were cut from the gel, and the RNA was extracted using the ZR small-RNA™ PAGE Recovery Kit (from Zymo Research). The RNA concentrations were also measured using a Nanodrop 8000 spectrophotometer (from Thermo Fisher).

**Toehold hairpin constructs preparation.** After agarose and PAGE gel purification, we measured and calculated the molarity of each component, including DNA handles, toehold hairpins, and adapter strands. Toehold hairpin strands and adapter strands are firstly mixed with a 1:1 molarity ratio of dT-Biotin-DNA handles (40 mM) and dT- Digoxigenin-DNA handles (40 mM), respectively. Next, the samples were dried using Concentrator 5301 eppendorf and resuspended in folding buffers (1 M NaCl, 50 mM HEPES, pH, 7.8 or 20 mM $MgCl_2$, 50 mM HEPES, pH 7.8). We incubated samples under different annealing temperature cycles (Table 1). We finally checked the folding constructs on an agarose gel.

**Preparation of the optical trap measurements and data analysis Measurement preparation.** To minimize multi-binding of the DNA handle, we diluted the fully hybridized constructs to a concentration of 0.4 nM and incubated them for 10 min with 1 μm-sized streptavidin-coated beads (Bangs Laboratories, Inc.) in 14 μL of running buffer (20 mM $MgCl_2$, 300 mM KCl, 50 mM HEPES, pH 7.2) at room temperature. In the meantime, we prepared the mobile phases in 500 μL of running buffer by adding an additional oxygen scavenger system (final concentrations: 26 U ml$^{-1}$ glucose oxidase (SIGMA-ALDRICH), 17 000 U ml$^{-1}$ catalase (SERVA), and 0.65 % glucose (SIGMA-ALDRICH).

The trigger phase included 0.1 μM of purified trigger strand, while the bead phase consisted of a mixture of streptavidin-coated beads and anti-digoxigenin-coated beads incubated together in 300 μL of the same scavenger system. All the phases were added to the syringe pump of the C-Trap® Optical Tweezers - Fluorescence & Label-free Microscopy (LUMICKS). We used a commercial microfluidic chip (LUMICKS) with multiple inlets that allowed us to generate a laminar flow to separate different phases (see Fig. 1B). In the bead channel, we trapped the two different kinds of beads, one in the fixed beam and the other one in the mobile beam. In the buffer channel, the molecular construct was tethered between the two beads by mobilizing one trap towards the fixed trap, resulting in the close proximity of the bead surfaces and the formation of a dumbbell-shaped conformation, known as the 'dumbbell assay' (See Fig. 1A). We maintained the laminar flow of buffer and trigger phase by applying ~ 0.35 bar to the syringes, leading to a flow velocity of ~ 20 μm s$^{-1}$ during the experiment to inhibit trigger diffusion into the buffer channel even for measurement times of up to an hour. In a typical experiment, designed to measure the invasion time, as shown in Fig. 2C, D, we initially established the tether between two beads and then transitioned to the trigger channel for the specific purpose of measuring trigger binding and strand invasion events exclusively. In all passive mode experiments, after binding the trigger strand in the trigger channel, we promptly relocated the beads back to the buffer channel to conduct pulling cycles and passive mode measurements. This step was taken to mitigate the potential influence of other trigger strands competing with the bound trigger strand and the unspecific binding of trigger strands to unfolded parts of the toehold hairpin. The trap stiffness used in all measurements was between 0.25 pN nm$^{-1}$ and 0.40 pN nm$^{-1}$. The sampling rate was 78.125 kHz and was down-sampled by a factor of 3 for data analysis except for the invasion time determination and autocorrelation analysis used for data shown in Fig. 2C, D. Measurement temperatures were ~ 25 °C.

**Data analysis.** We used Igor Pro 8.0 (Wavemetrics, Lake Oswego, OR) to analyze the force-extension curves obtained from the stretch and relax cycles and the passive mode data. For the force-extension traces, we employed a worm-like chain (WLC) model as described in detail in the Supporting Information (Supplementary Methods "Stretch and relax cycles". HMM-based assignment of states was performed to assign SMFS data acquired in passive mode experiments to intermediate states[48], and kinetics and energetics were thereafter extracted as shown in detail in the Supporting Information (Supplementary Methods "Passive mode traces", "Extracting free energies", "Model for transition-rate extrapolation" and "Invasion time determination and autocorrelation analysis").

## OxDNA simulations

Kinetic simulations were performed on the full system containing linker strands, hairpin and invader. Free energy simulations were performed on a shortened system. Details are described in depth in the Supporting Information (Supplementary Methods "Simulation protocols").

## Reporting summary

Further information on research design is available in the Nature Portfolio Reporting Summary linked to this article.

## Data availability

The authors declare that the SMFS data supporting the findings of this study are available within the paper, its Supplementary Information file, the Source Data file, and FigShare at https://doi.org/10.6084/m9.figshare.26474698. SMFS raw output data from Bluelake 2.4.2 (Lumicks) with analysis instructions using licensed software Igor Pro 8.0 (Wavemetrics) are available from the corresponding authors upon

request. Simulation data supporting the findings of this study are available within the paper, and its Supplementary Information, and the setup required to reproduce all datasets is available at https://github.com/zoombya/SMFS_TMSD.

## Code availability

The simulation source code is freely available under GNU Public License, along with documentation, installation instructions, and example datasets at https://github.com/lorenzo-rovigatti/oxDNA. The simulation code configurations and setup required to reproduce the datasets studied in this work are available at https://github.com/zoombya/SMFS_TMSD.

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

## Acknowledgements

This work was supported by the Deutsche Forschungsgemeinschaft through SFB 863 (project ID 111166240 TPA2 (to M.R.) and TPA8 (to F.C.S.)). This result is part of a project that has received funding from the European Research Council (ERC) under the European Union's Horizon 2020 research and innovation program (Grant Agreement No. 101040035) (to P.Š.). We appreciate the help of M.Sc. Andreas Weißl who helped during the early phases of the project.

## Author contributions

A.W., M.R., T.W., F.C.S., conceived the project. A.W. and T.W. designed the sequences and performed the experiments. T.W. synthesized the constructs used for SMFS experiments. A.W. analyzed the SMFS data. M.M. and P.S. performed simulations. P.S., F.C.S., and M.R. supervised the project. All authors co-wrote the paper.

## Funding

## Competing interests

The authors declare no competing interests.
