## [Peer Review File · Nature Communications]

Single-Molecule Force Spectroscopy of Toehold-Mediated Strand DisplacementREVIEWER COMMENTS

Reviewer #1 (Remarks to the Author):

In this manuscript, Walbrun et al. presented single-molecule force spectroscopic data of the toehold-mediated strand displacement (TMSD) process for different nucleic acid invader/target combinations (DNA/DNA, RNA/RNA, DNA/RNA). The experiments are complemented with coarse-grained simulations using an oxDNA model. The results depict the branch migration dynamics under load—a physiologically relevant scenario—with microsecond and nanometer resolution.

This is a well-executed study with high-quality data that maintain the usual standard of the research team. Although some of the findings are confirmatory, the force- and sequence-dependent quantitative results still provide a valuable contribution to the field and will help the rational design of riboregulators for nanotechnology and synthetic biology applications. Below are a few points that the authors should address.

- (1) The invasion times were obtained differently for DNA (exponential fitting) and RNA (visual inspection). The longer transition time for RNA was attributed to a quasi-stable intermediate state. However, it would still be interesting to know whether the stepping for RNA branch migration per se is slower than that for DNA branch migration. Can the authors do exponential fitting to the transition segments before and after the intermediate, or adjust the sequence to remove the intermediate state?
- (2) Is it possible to simulate the RD invasion scenario to support the experimental data in Fig. 5?
- (3) In Discussion (page 13), the authors mentioned that their experimental design “allows the determination of position-independent displacement kinetics”. Nonetheless, the position of mismatches can still have an impact on strand displacement, which this study does not address. Curiously, DDp1 and DDp2 constructs are shown in Fig. S1 but never mentioned in the results.
- (4) The term “re-invasion” is confusing. It describes the reverse process of strand invasion but could easily be confused with a repeat of the invasion step.
- (5) On page 5: “We find that the invasion times are largely independent of force (Fig. 2F, Fig. S5)...” Fig. 2F doesn’t exist.
- (6) On page 5: “Such a mismatch is expected to retard the transition between the toehold-bound (TB) and the fully invaded (FI) state by imposing a free-energy penalty on invasion while leaving re-invasion unaffected.” I don’t quite follow the logic here. The effect of mismatch on invasion/re-invasion rates should depend on where the transition state is. If anything, the FI state should be destabilized by the mismatch.
- (7) How is the energy landscape shown in Fig. 5D different from the one in Fig. S8G? Is it a coincidence that the intermediates (RD1, RD2, RD3) are all roughly equidistant?
- (8) For several conditions only 1 molecule was examined (for example, 4th paragraph on page 6). It is unclear what “N” means in this case. Each condition should be tested on at least a couple of molecules to ensure consistency.
- (9) The supplemental materials are extensive. Some of them are quite important and can be moved to the main text/figures (e.g., Fig. S4, S8).

Reviewer #2 (Remarks to the Author):

This work explored the dynamics of strand displacement processes at the single-molecule level using single-molecule force spectroscopy (SMFS) with an optical trap supported by state-of-the-art coarse-grained simulations. The equilibrium state and energy landscape within TMSD were resolved through SMFS and oxDNA simulation. Under the influence of forces of varying magnitudes, the invasiveness exhibited varying degrees of enhancement. Dynamical analysis of the invasion process revealed that the single-step time for DNA invading DNA is four times that of RNA invading RNA. Additionally, the kinetics of DNA invading RNA, a non-spontaneous reaction, was investigated.

In general, this is a novel research method. As for simulation and corresponding analysis, some issues should be explained and clarified as follows:

1. In the Figure 4A and B, the force-induced unfolding process of the remaining stem after the trigger invasion domain should be the same, please explain why the curves at 22 pN differ in the two figures.
2. As the Figure 4C depicting, under the force of 5 pN, the trend of free energy change in the branch migration is consistent with that toehold binding. Can it be considered that 5 pN has completely destroyed the binding of incumbent? Or what would the free energy change be like without incumbent?
3. In the Figure 4D, it is an interesting point that how large force could counteract the energy barrier due to mismatch. Additionally, no base pair can be formed at the mismatch, which is better represented by leaving the corresponding site blank in the figure.
4. During the above simulation, it is noted that an extra adenine was added to the 3' end to pull force. Please further explain for this.
5. In the study of DNA invading RNA, sequence effects were mentioned. However, comparison of different sequences was not addressed in the main text.
6. In the TMSD system with mismatch, the displacement after mismatch site should be the same. As for Figure S11D and E, under the constant force, please further explain why the free energy of the branch migration process had been decreasing and with different trend?
7. We noticed that the Figure 2F was mentioned in the last paragraph of "Single-molecule observation of toehold-mediated strand displacement (TMSD)", but there was no corresponding figure. Please check and correct.

Manuscript NCOMMS-24-07769-T

MS Type: Research Article

Title: "Single-Molecule Force Spectroscopy of Toehold-Mediated Strand Displacement"

Corresponding Authors: Matthias Rief and Friedrich Simmel

Detailed response to the Referee comments:

Reviewer Comments:

Reviewer #1:

In this manuscript, Walbrun et al. presented single-molecule force spectroscopic data of the toehold-mediated strand displacement (TMSD) process for different nucleic acid invader/target combinations (DNA/DNA, RNA/RNA, DNA/RNA). The experiments are complemented with coarse-grained simulations using an oxDNA model. The results depict the branch migration dynamics under load—a physiologically relevant scenario—with microsecond and nanometer resolution.

This is a well-executed study with high-quality data that maintain the usual standard of the research team. Although some of the findings are confirmatory, the force- and sequence-dependent quantitative results still provide a valuable contribution to the field and will help the rational design of riboregulators for nanotechnology and synthetic biology applications. Below are a few points that the authors should address.

We are glad to hear the positive feedback by this reviewer.

(1) The invasion times were obtained differently for DNA (exponential fitting) and RNA (visual inspection). The longer transition time for RNA was attributed to a quasi-stable intermediate state. However, it would still be interesting to know whether the stepping for RNA branch migration per se is slower than that for DNA branch migration. Can the authors do exponential fitting to the transition segments before and after the intermediate, or adjust the sequence to remove the intermediate state?

We thank the reviewer for this suggestion. In fact, we were originally a bit skeptical about fitting the exponential trace to a subpart of the relaxation trace because we were worried about artefacts due to the choice of the starting points for those fits when the start and end levels are closer together and the noise bands of both levels would overlap. However, we followed the suggestion and are really happy with the results. As suggested by the reviewer we analyzed just the fast parts of the relaxation using an exponential fit. Only the portion after the intermediate could be reliably fit because for the portion before, the start and end levels were too close. We fitted an exponential function using the same method as for the system DD in Fig. 2C (see "Invasion time determination and autocorrelation analysis" for details). From this analysis we get an average invasion time of $213 \pm 17 \mu\text{s}$ at an average force of 13.6 pN for RNA invading RNA. This value is 5.1 times slower compared to DNA ($42 \pm 5 \mu\text{s}$) at an average force of 9.8 pN. Interestingly, this factor is very close to the factor of 4.4 we find between single step times for RNA compared to DNA at zero load extracted from the equilibrium measurements in Fig. 3C and D. Even though the value we found for DNA using the exponential fit was very close to the system response time the difference between DNA and RNA is significant. That's why we decided to include the values obtained for RNA also into Fig. 2D. Interestingly a validation of the fitting procedure for RNA came for free since we also included some data measured in absence of Mg ions (light green squares in Fig. 2D). In this case the fitting procedure yielded significantly faster values despite identical starting and end points from intermediate to FI. We feel that now, from the comparison of the values we find both under load (Fig. 2) as well as in the absence of load (Fig. 3), a consistent picture of the differences in branch migration step times emerges between RNA and DNA. We adapted results and discussion sections accordingly.

(2) Is it possible to simulate the RD invasion scenario to support the experimental data in Fig. 5?

At the time of submission, a DNA-RNA coarse-grained model was not available. Recently our colleagues from Oxford University have published a new implementation that combines oxDNA and oxRNA models into a hybrid model implemented only on the CPU. To accompany the experimental data in Fig 5 we would need a GPU version of the hybrid DNA-RNA model, which is not implemented yet. While the Oxford group plans to port the DNA-RNA hybrid model to GPU eventually, there is no time estimate when the new GPU version will be available. Until then simulations of invasion kinetics of the size of the experimental system will be out of reach.

(3) In Discussion (page 13), the authors mentioned that their experimental design “allows the determination of position-independent displacement kinetics”. Nonetheless, the position of mismatches can still have an impact on strand displacement, which this study does not address. Curiously, DDp1 and DDp2 constructs are shown in Fig. S1 but never mentioned in the results.

It is correct that the position of mismatches plays a significant role in the strand displacement process which was studied extensively in previous works^{1,2}. Here, we mainly focused on determining the kinetics of a single branch migration step in contrast to the total strand displacement times that include binding of the invader from the solution to the toehold. We wanted to highlight that in the system we use in our optical tweezers experiments we would not expect that the position of the mismatch affects the reported rates as neither spontaneous dissociation¹ can occur in our system nor short toehold lengths were used.

We changed the phrasing (page 13) “Our design is independent of these effects and allows the determination of position-independent displacement kinetics for two reasons” to “Our design is independent of these effects and different positions of the mismatch are not expected to affect the invasion rate once the trigger strand is bound to the toehold for two main reasons.” where we removed “displacement kinetics” which is ambiguous and could also be understood as the total time it needs to bind to the toehold and displace the incumbent strand. We hope that this makes our statement clearer.

For the simulations, we actually tested proximal as well as central mismatches. However, the constructs DDp1 and DDp2 in Fig. S1 that show the sequence of the proximal mismatches for the DNA case are the sequences for the full system and not the simplified system where we reduced the length of both toehold hairpin and trigger strand as the full system is very compute-intensive. We corrected this mistake and also added RD and the RRp2 system that was used in Fig. 2 and Fig. S3 and S6.

(4) The term “re-invasion” is confusing. It describes the reverse process of strand invasion but could easily be confused with a repeat of the invasion step.

Thank you for pointing that out. We replaced all the “invasion/re-invasion” terms to “forward/backward invasion”.

(5) On page 5: “We find that the invasion times are largely independent of force (Fig. 2F, Fig. S5)...” Fig. 2F doesn't exist.

Thank you for addressing this error. We corrected it to “Fig. 2D”.

(6) On page 5: “Such a mismatch is expected to retard the transition between the toehold-bound (TB) and the fully invaded (FI) state by imposing a free-energy penalty on invasion while leaving re-invasion unaffected.” I don't quite follow the logic here. The effect of mismatch on invasion/re-invasion rates should depend on where the transition state is. If anything, the FI state should be destabilized by the mismatch.

We agree with the reviewer that this sentence implies assumptions which do not apply generally. We reformulated the sentence as follows: Such a mismatch will raise the free energy of the fully invaded (FI) state over the toehold-bound (TB) state because it possesses one complementary base pair less. Application of force will now introduce an additional intermediate state at the mismatch position (IM). The force can now be chosen such that the IM state and the FI state have the same free energy and continuous forward/backward invasion between IM and FI will be observed....

(7) How is the energy landscape shown in Fig. 5D different from the one in Fig. S8G? Is it a coincidence that the intermediates (RD1, RD2, RD3) are all roughly equidistant?

The free energy landscape in Fig. 5D (see below left) is obtained from the experiment (gray: at the measured average force, black: extrapolated to zero load), while the landscape in Fig. S8G (see below right) is an extrapolated landscape at the force of the experiment from the simple nearest neighbor model landscape in Fig. S8F (see below middle). We can compare now the landscape in Fig. S8G with the gray landscape in Fig. 5D and the main difference we see is that the experimental landscape shows higher barriers and the position of the minima do also not completely overlap. We propose that elaborate models need to be developed to understand this sequence dependence.

Why the intermediate positions are spaced equidistantly may be coincidence. Interestingly, also the nearest neighbor model despite falling short in terms of barrier heights still predicts slight minima at the position RD1 and RD3. We need further theoretical modeling to fully understand this effect.

(8) For several conditions only 1 molecule was examined (for example, 4th paragraph on page 6). It is unclear what “N” means in this case. Each condition should be tested on at least a couple of molecules to ensure consistency.

We totally agree that it is important that each condition must be tested on multiple molecules to ensure reproducibility and enhance statistical power. In the special case of the 4th paragraph on page 6, we describe Fig. 3D where we measure the rate dependence of forward and backward invasion as a function of load. In such a case, we prefer to display the data for just one molecule because it avoids mixing systematic errors of force calibration of multiple molecules. The single molecule data in fact serves as a quality control because the reader can convince themselves that the molecule showed a nice linear behavior over a large force range without deteriorating or getting damaged. If we pieced together such a graph from subsets of many molecules the lines of each individual molecule would be slightly shifted with respect to each other due to the systematic errors between the different measurements thus complicating the analysis. To illustrate this, we have made such an overlay for DDc1 in the graph below. When extracting average values for the zero force rates as well as transition state positions we use averages of many molecules as shown in the last paragraph of page 6 “weighted mean: $780 \pm 30 \text{ s}^{-1}$ (DDc1, N = 102, 9 molecules, s.e.m., see Tab. S5)”.

Figure. Direct observation of repeated forward/backward invasion in the system DDc1. All molecules measured are shown compared to the representative molecule in Fig. 3D. Black symbols: forward invasion. Gray symbols: backward invasion. The zoom enhances visibility of different symbols representing data points from different molecules. The deviation or spread between different molecules is explained in the section above. M1 bw stands

for molecule 1 backward invasion, M1 fw for molecule 1 forward invasion and the other molecules are termed analogously.

In the case of Fig. 2D, we originally also showed data from a single molecule and we have now included additional data from more molecules.

There are some analyses especially in the tables of the SI where we extract lengths as well as free energy values where the full analysis was performed for only one molecule, due to a time-consuming and elaborate analysis. For example, the free-energy measurements in Table S2 requires an elaborate passive mode analysis at various different forces over many minutes. It can be ensured that the molecule chosen exhibits a representative behavior by first recording simple stretch-relax cycles that already show that the molecule has the right length, the right intermediate states as well as the right unfolding forces as well as overall kinetics. We have measured many more molecules that we can compare this behavior to. In the graph below, we show a stretch relax cycle of the molecule that went into the analysis of Tab. S2 (first trace from the left) together with 10 other molecules measured on other days. We did this kind of consistency check with every molecule that went into a “single molecule” analysis.

Figure. Stretch (black) and relax (gray) cycles of the DNA hairpin. The trace on the left was used in the free energy calculation (see Tab. S2). The other traces are from different DNA hairpin molecules and show consistent unfolding patterns verifying reproducibility.

(9) The supplemental materials are extensive. Some of them are quite important and can be moved to the main text/figures (e.g., Fig. S4, S8).

We are glad the reviewer acknowledges the importance and information content of our supplemental materials. Following the reviewer’s suggestion to do an additional analysis of the branch migration kinetics in RNA in Fig. 2, we already decided to include this data into the main text. In general, we tried to use an approach where we show a representative trace in the main figure and then a summary of extracted values from such a trace (see Fig. 2C (representative), Fig. 2D (summary), Fig. 3C (representative), Fig. 3D (summary)). We show more data traces in Fig. S4 and S7 to provide the reader with a better understanding about the statistical variations between individual sample traces but it does not give much additional information for relevant parameters that are used later or in the discussion and we therefore decided to put them into the SI.

Reviewer #2:

This work explored the dynamics of strand displacement processes at the single-molecule level using single-molecule force spectroscopy (SMFS) with an optical trap supported by state-of-the-art coarse-grained simulations. The equilibrium state and energy landscape within TMSD were resolved through SMFS and oxDNA simulation. Under the influence of forces of varying magnitudes, the invasiveness exhibited varying degrees of enhancement. Dynamical analysis of the invasion process revealed that the single-step time for DNA invading DNA is four times that of RNA invading RNA. Additionally, the kinetics of DNA invading RNA, a non-spontaneous reaction, was investigated.

In general, this is a novel research method. As for simulation and corresponding analysis, some issues should be explained and clarified as follows:

We thank the reviewer for this positive reply and hope to solve the issues in the following.

(1) In the Figure 4A and B, the force-induced unfolding process of the remaining stem after the trigger invasion domain should be the same, please explain why the curves at 22 pN differ in the two figures.

Assuming that the reviewer meant Figure 4A left and right panel: we combined them for better visualization below. The two systems are quite different, as in the no trigger case the unfolded hairpin is single-stranded, and in the case of the DNA/DNA invader the invaded part is double-stranded. As can be seen below, the process for the hairpin invasion is quicker than the hairpin unzipping by the pulling, therefore the DD curve has a different slope than the “no trigger” curve.

It has been shown previously that in the oxDNA equilibrium simulations of a smaller hairpin system, the hairpin unzipping happens at about 20 pN (<https://pubs.acs.org/doi/full/10.1021/acsnano.5b04726>). As the system in our simulation is pulled (and hence out of equilibrium) with a relatively high speed the estimated force value is shifted upwards. In the case with the invader strand (labeled as DD in the figure below), after the strand fully invades, the applied pulling force is about at 25 pN, enough to quickly open the remaining 16 bp hairpin, thus we do not observe a plateau as in the case where the longer (52 base pairs) hairpin is unzipped without the help of the invader. If the remaining hairpin after invasion would be much larger than 16 base pairs, we would expect the plateau at larger force to also appear in the force-extension curve. We added this version of the overlaid force-extension curves, where the focus is on the comparison of the two systems and where the effect of opened extension between single- and double-strands is clearer, to the SI (Fig. S12A).

(2) As the Figure 4C depicting, under the force of 5 pN, the trend of free energy change in the branch migration is consistent with that toehold binding. Can it be considered that 5 pN has completely destroyed the binding of incumbent? Or what would the free energy change be like without incumbent?

The figure 4C shows a projection of total free energy as a function of the number of base pairs between the invader and the incumbent. A more fine-grained picture is provided in the supplement (Fig. S11), where we show a 2D plot of the free energy as a function of the invader's base pairs and incumbent's base pairs. The free energy simulation samples different states in terms of the number of base pairs formed, and shows it is more favorable for the system to invade and replace the incumbent under the tension. However, tracing out the minimum free energy pathway through the landscape, it favors a case where one broken base pair of incumbent is exchanged with a new base pair with the invader. This is in line with our kinetic pulling simulations on GPUs, where we saw that the invader invades base by base. We added an explanation to the SI (see figure caption of Fig. S11).

(3) In the Figure 4D, it is an interesting point that how large force could counteract the energy barrier due to mismatch. Additionally, no base pair can be formed at the mismatch, which is better represented by leaving the corresponding site blank in the figure.

We thank the reviewer for pointing this out, as that was one of the effects that we aimed to communicate with our free-energy plots. The plots are shown in terms of the number of base pairs formed between the incumbent and the invader, so we would find it a bit difficult to introduce a "mismatch" site on the x-axis, as there would be no free energy that would be assigned to a base pair in that state.

(4) During the above simulation, it is noted that an extra adenine was added to the 3' end to pull force. Please further explain for this.

We simulated a shortened system, which is just a subset of the strands used in the experimental system. In the experiment, the forces are applied on the handles, so in order to get a more comparable scenario of force transmission to the branch migration junction we added one extra base (A) to the incumbent, so that the force is applied on this extra base rather than directly on the base in the first base pair in the incumbent. On the 5'-side of the substrate strand this was not necessary as the crucial part is the branch migration junction and the 5'-end is at least 10 bases away from this junction. We now included a paragraph where we explain this to the SI before Fig. S11.

(5) In the study of DNA invading RNA, sequence effects were mentioned. However, comparison of different sequences was not addressed in the main text.

We agree with the reviewer that the topic of sequence effects deserves further study. The purpose of the present study is the comparison of branch migration on one model sequence using different combinations of RNA and DNA. Already, this one sequence contains a large set of different nearest neighbor combinations, which allowed us to conclude that sequence variations are important to understand branch migration in DNA/RNA hybrids. However, we feel that a full understanding of these effects warrants a future study in its own right, particularly with the new simulation methods that will hopefully be available soon for systems of the size we study (see answer to question 2, reviewer 1).

(6) In the TMSD system with mismatch, the displacement after mismatch site should be the same. As for Figure S11D and E, under the constant force, please further explain why the free energy of the branch migration process had been decreasing and with different trend?

Different colors correspond to the different pulling forces. The slope is different for different forces, as higher pulling force makes it more favorable for the invader to displace the incumbent, which is under tension. We aligned the free energy profiles to be equal to 0 when the toehold is formed, which might create an impression that the free-energy is the same until the mismatch, as the mismatches are right after the toehold.

(7) We noticed that the Figure 2F was mentioned in the last paragraph of “Single-molecule observation of toehold-mediated strand displacement (TMSD)”, but there was no corresponding figure. Please check and correct.

Thank you for addressing this error. We corrected it to “Fig. 2D”.

References

- 1 Irmisch, P., Ouldrige, T. E. & Seidel, R. Modeling DNA-strand displacement reactions in the presence of base-pair mismatches. *Journal of the American Chemical Society* **142**, 11451-11463 (2020).
- 2 Machinek, R. R., Ouldrige, T. E., Haley, N. E., Bath, J. & Turberfield, A. J. Programmable energy landscapes for kinetic control of DNA strand displacement. *Nature communications* **5**, 5324 (2014).

REVIEWERS' COMMENTS

Reviewer #1 (Remarks to the Author):

The authors did an excellent job addressing my questions and the manuscript is significantly improved. I therefore enthusiastically recommend its publication.

Reviewer #2 (Remarks to the Author):

The authors have addressed my concerns